# Structure of SARS-CoV-2 membrane protein essential for virus assembly

Zhikuan Zhang[1], Norimichi Nomura [2], Yukiko Muramoto [3,4,5], Toru Ekimoto[6], Tomoko Uemura[2], Kehong Liu[2], Moeko Yui[1], Nozomu Kono [1], Junken Aoki[1], Mitsunori Ikeguchi[6,7], Takeshi Noda[3,4,5], So Iwata[2,8], Umeharu Ohto [1] ✉ & Toshiyuki Shimizu [1] ✉

The coronavirus membrane protein (M) is the most abundant viral structural protein and plays a central role in virus assembly and morphogenesis. However, the process of M protein-driven virus assembly are largely unknown. Here, we report the cryo-electron microscopy structure of the SARS-CoV-2 M protein in two different conformations. M protein forms a mushroom-shaped dimer, composed of two transmembrane domain-swapped three-helix bundles and two intravirion domains. M protein further assembles into higher-order oligomers. A highly conserved hinge region is key for conformational changes. The M protein dimer is unexpectedly similar to SARS-CoV-2 ORF3a, a viral ion channel. Moreover, the interaction analyses of M protein with nucleocapsid protein (N) and RNA suggest that the M protein mediates the concerted recruitment of these components through the positively charged intravirion domain. Our data shed light on the M protein-driven virus assembly mechanism and provide a structural basis for therapeutic intervention targeting M protein.

Severe acute respiratory syndrome coronavirus 2 (SARS-CoV-2) has caused the ongoing pandemic of 2019 coronavirus disease (COVID-19), which has taken the lives of countless people[1,2]. SARS-CoV-2 belongs to the genus *Betacoronavirus*, which mainly consists of four structural proteins: spike (S) protein, nucleocapsid (N) protein, envelope (E) protein, and membrane (M) protein (Fig. 1a)[1,3]. The S protein is responsible for binding to the host entry receptor angiotensin-converting enzyme-2 (ACE-2) and cell membrane fusion[4,5], N protein for packaging of single-stranded RNA genome (ssRNA)[6,7] and E for ion conduction as a viral ion channel (viroporin)[8]. M protein is the most abundant viral structural protein and is considered the major driver of virus assembly and membrane budding[9,10].

M protein synthesized in the host cells is localized in the endoplasmic reticulum–Golgi intermediate compartment (ERGIC) and provides a platform for recruiting other viral structural proteins[10]. Co-expression of SARS-CoV-2 structural proteins in mammalian cells generates virus-like particles (VLPs), where a minimal combination of M + N is required for VLP formation[11–13]. This indicates the indispensable role of M protein in virus assembly. The species-specific pairing of M and N proteins is essential for the correct formation of virus particles in SARS-CoV and murine coronavirus mouse hepatitis virus (MHV)[14]. In addition, the interactions between M protein and N protein of SARS-CoV-2[15] and between M protein and genome RNA containing a packaging signal of MHV[16] have been reported. These data

[1]Graduate School of Pharmaceutical Sciences, The University of Tokyo, 7-3-1 Hongo, Bunkyo-ku, Tokyo, Japan. [2]Department of Cell Biology, Graduate School of Medicine, Kyoto University, Yoshida Konoe-cho, Sakyo-ku, Kyoto, Japan. [3]Laboratory of Ultrastructural Virology, Institute for Life and Medical Sciences, Kyoto University, 53 Shogoin Kawahara-cho, Sakyo-ku, Kyoto, Japan. [4]Laboratory of Ultrastructural Virology, Graduate School of Biostudies, Kyoto University, 53 Shogoin Kawahara-cho, Sakyo-ku, Kyoto, Japan. [5]CREST, Japan Science and Technology Agency, 4-1-8 Honcho, Kawaguchi, Saitama, Japan. [6]Computational Life Science Laboratory, Graduate School of Medical Life Science, Yokohama City University, 1-7-29, Suehiro-cho, Tsurumi-ku, Yokohama, Kanagawa, Japan. [7]HPC- and AI-driven Drug Development Platform Division, Center for Computational Science, RIKEN, Yokohama, Japan. [8]RIKEN SPring-8 Center, Kouto, Sayo-cho, Sayo-gun, Hyogo, Japan. ✉e-mail: umeji@mol.f.u-tokyo.ac.jp; shimizu@mol.f.u-tokyo.ac.jp

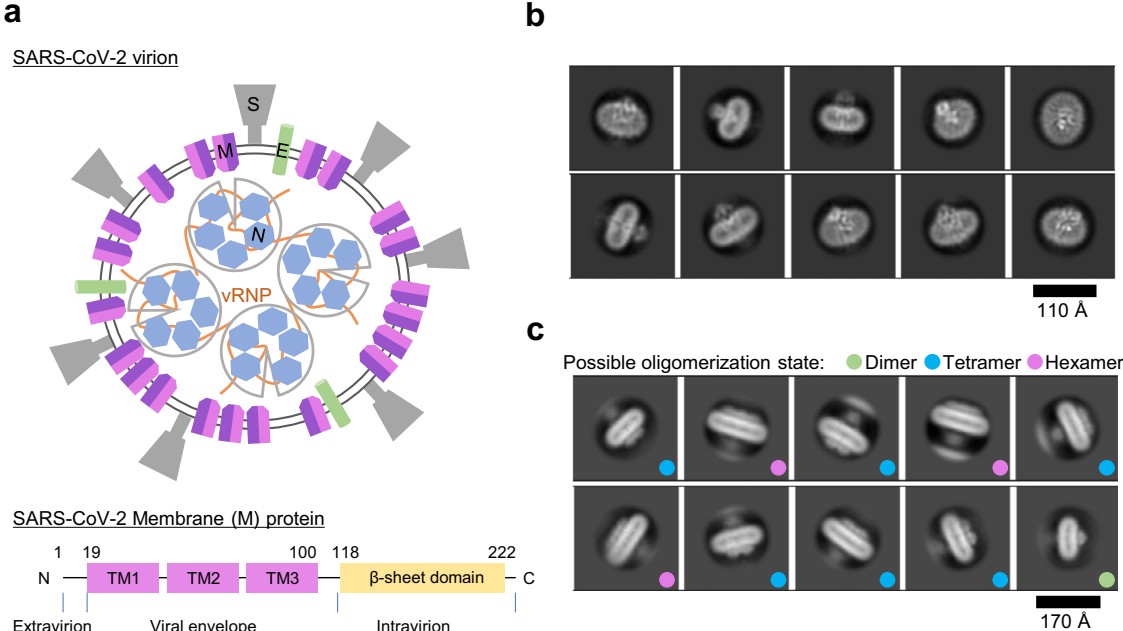

**Fig. 1 | M protein forms dimer and higher-order oligomers. a** Schematic view of SARS-CoV-2 virion. Four structural proteins, M, N, S, and E and genomic RNA are shown in different colors. The N protein and RNA molecules form viral ribonucleoprotein (vRNP) complexes. **b** Representative 2D class average images of M protein dimer solubilized by LMNG-CHS (see also Supplementary Fig. 2) **c** Representative 2D class average images of higher-order oligomers of M protein solubilized by GDN (see also Supplementary Fig. 3). The possible oligomerization state is indicated by the colored dots.

in SARS-CoV-2 and related viruses suggest the versatile and vital role of M protein in coronavirus assembly.

M protein contains three transmembrane helices and a C-terminal intravirion domain (Fig. 1a). Phylogenetic analysis revealed that the *Betacoronavirus* M protein family is evolutionarily related to ORF3a in *Sarbecovirus*, ORF5 in Middle East Respiratory Syndrome (MERS) CoV, and M proteins from more distant toroviruses[17,18]. The recently reported cryo-EM structures of SARS-CoV-2 ORF3a revealed that ORF3a mainly forms dimers and functions as a nonselective cation-permeable viroporin[19]. These data suggest that the SARS-CoV-2 M protein also functions as a viroporin.

Despite its critical importance for virus assembly, the precise role of the M protein in this process remains largely unknown. Here, we report the cryo-EM structures of M protein in two distinct conformational states and M/N/RNA interaction analysis that provide a clue to understanding the mechanism underlying an essential step in virus assembly.

## Results

### SARS-CoV-2 M protein forms dimer and higher-order oligomers

To prepare the SARS-CoV-2 M protein for structural determination, we examined the expression of M protein using *Escherichia coli*, baculovirus, and HEK293 expression systems. Among them, we selected HEK293 cells to purify recombinant full-length M protein (see Methods). The M protein solubilized by either LMNG/CHS or GDN detergents yielded similar monodisperse peaks in gel filtration purification (Supplementary Fig. 1a, b). GDN-solubilized M protein also contained a small portion (<10%) of M protein in a higher oligomeric state (Supplementary Fig. 1b). We analysed the LMNG/CHS solubilized M protein using cryo-electron microscopy (cryo-EM) (Supplementary Fig. 2). The two-dimensional (2D) class averages and low-resolution 3D reconstruction showed clear structural features with C2 symmetry, composed of a compact intravirion domain and transmembrane helices, suggesting that the M protein formed a dimer (Fig. 1b). Additionally, we examined the GDN-solubilized M protein in a higher oligomeric state using cryo-EM (Supplementary Fig. 1b, 3). The 2D class averages

showed laterally aligned M protein dimers possibly through interactions at the intravirion domains, which corresponded to tetrameric and even hexameric M proteins in addition to dimers (Fig. 1c). These results are consistent with the results of a previous cryo-EM tomographic study of mouse hepatitis virus (MHV) M protein inside virions, demonstrating that the M protein forms oligomers[20].

### Structure determination of SARS-CoV-2 M protein

To obtain high-resolution structural information about the M protein, we generated mouse monoclonal anti-M protein antibodies and utilized their Fab fragment as a fiducial marker in cryo-EM analysis (see Methods). We chose two Fab fragments (YN7756_1, Fab-E; YN7717_9, Fab-B) among the six Fab fragments tested (Supplementary Fig. 1c), and successfully determined the structures of the M/Fab-E and M/Fab-B complexes at 2.7- and 2.8-Å resolution, respectively (Fig. 2a and Supplementary Figs. 4–7). In both cases, the M protein adopted a C2 symmetric dimeric structure with two Fab fragments bound to the intravirion region of the M protein (Fig. 2a). Surprisingly, M protein dimers in the M/Fab-E and M/Fab-B complexes differed in their conformations, with sizes of 86-Å height × 50-Å width (long form) or 72-Å height × 57-Å width (short form), respectively (Fig. 2b). Considering that antibodies used in the cryo-EM analysis were raised against M protein, it is reasonable to assume that the M protein is in a conformational equilibrium between the two states and that each state was captured by its conformation-specific antibody. In line with this idea, the long and short forms of M protein inside MHV virions have also been observed in a previous cryo-EM tomographic study[20]. Interestingly, the long-form structure fitted well into the low-resolution cryo-EM reconstruction of the Fab-unbound M protein dimer (Supplementary Fig. 8).

The M protein consists of three structural segments: the N-terminal three transmembrane helices (residues 9–105) mostly embedded in the detergent micelle (i.e., the viral envelope), a juxtamembrane hinge region (residues 106–116), and an inward-facing C-terminal β-sheet sandwich domain (BD) (residues 117–201) composed of an outer sheet (β1, β2, β6, and β7) and an inner sheet (β3, β4, β5, and

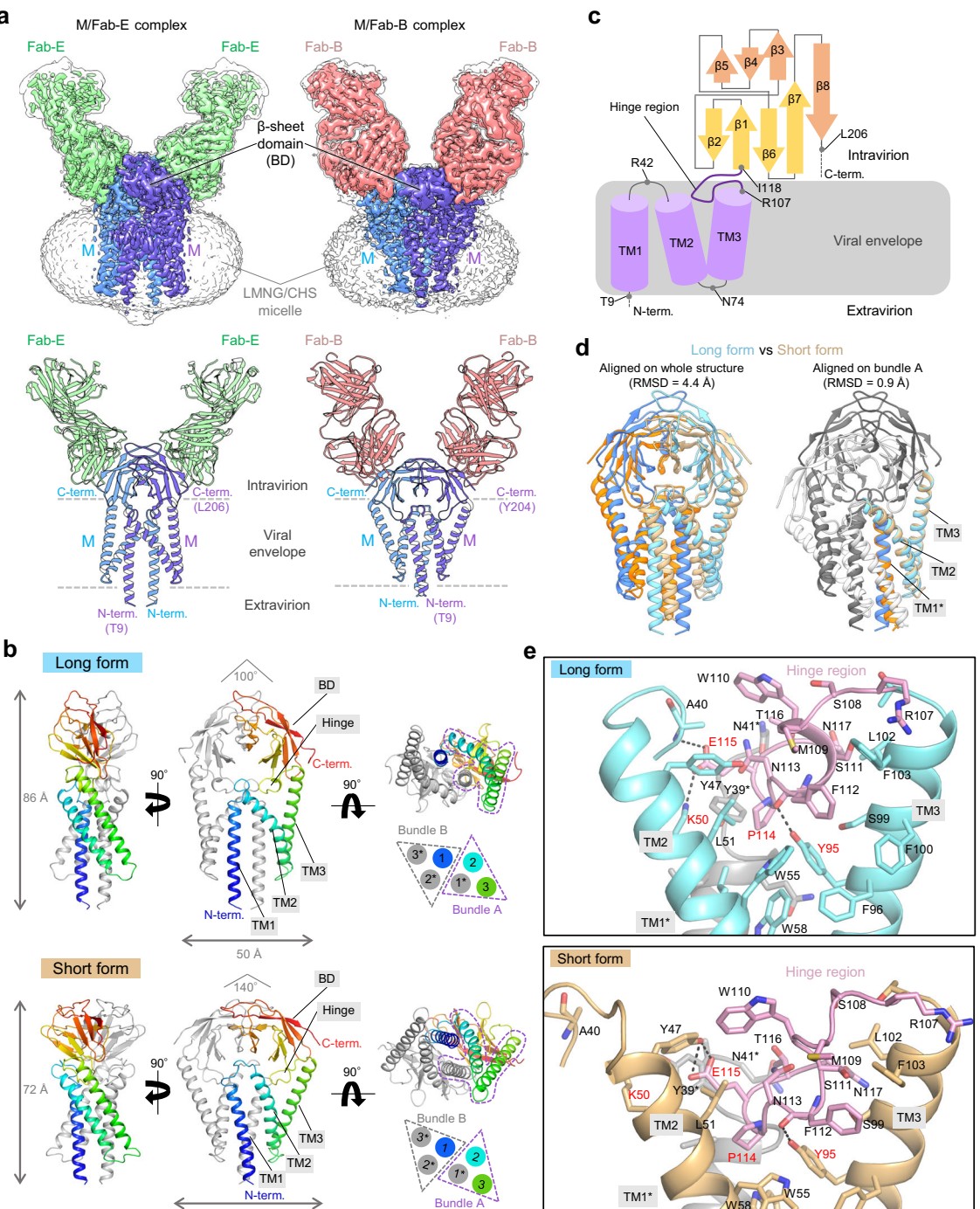

**Fig. 2 | Cryo-EM structures of M protein dimer. a** Cryo-EM maps (upper) and ribbon models (lower) of M/Fab-E and M/Fab-B complexes. Unsharpened maps are shown in transparent light gray color (M/Fab-E complex: level = 0.08; M/Fab-B complex: level = 0.35), and B-factor-sharpened maps are show in multiple colors (M/Fab-E complex: level = 0.28; M/Fab-B complex: level = 0.75). M protein protomers, Fab-E, and Fab-B, are colored in purple, blue, green, and pink, respectively. **b** Structures of M protein dimer in the long form (upper) and the short form (lower) are shown from the side (left), front (middle), and bottom (right) views. One protomer of M protein is colored in a rainbow, and the other is light gray. Schematic views of the arrangement of six TM helices are shown on the lower right. TM1*, TM2,

and TM3 formed one bundle. Throughout this paper, asterisks are used to indicate structural parts from the second M protein protomer. **c** Schematic view of the M protein protomer. Secondary structure elements and hinge regions are indicated. The disordered N-term and C-term regions are indicated by dashed lines. **d** Structure comparison between the long form (blue and cyan) and short form (orange and beige). The structures were aligned either according to the whole structure (left) or bundle A (right). **e** Detailed views of the interactions between the hinge region and bundle A in the long (upper) and short (lower) forms. The key residues are indicated in red. Dashed lines represent hydrogen bonds or salt bridges.

β8) (Fig. 2a–c and Supplementary Fig. 6). The N-terminal region (residues 1–8, extravirion) and C-terminal regions (residues 207–222, intravirion) of the M protein are disordered. The dimerization of the M protein is mediated by both transmembrane helices and the BD

domain. In both forms of the M protein dimer, TM1 in one protomer is domain-swapped to form a three-helix bundle with TM2 and TM3 in the other protomer (Fig. 2b). The interactions within a single three-helix bundle are mediated by extensive contact between hydrophobic

residues (Supplementary Fig. 9a). The hinge region adopts a helix-turn structure and is inserted into a triangular pyramidal cavity formed by the three-helix bundle (Fig. 2b). In addition, the BD dimerizes through the inner sheet to form an umbrella shape that caps the intravirion face of the transmembrane region (Fig. 2b, e).

## M protein adopts different conformational states

Whereas the conformations of each bundle (bundles A and B) and each BD remained largely unchanged between the two forms of M protein dimer, the relative arrangements between bundles A and B and between the two BDs were different (Fig. 2d and Supplementary Movie 1). When transitioning from the long form to the short form, the two bundles rotate toward each other on the extravirion side and against each other on the intravirion side. Consequently, the linker between TM1 and TM2 is widely twisted outward from the dimerization axis, and the interface between the two bundles is rearranged so that the two bundles are closer on the extravirion side and more separated on the intravirion side. Synchronized with the motion of the two bundles, the dimerization interface at BDs is also largely rearranged so that the BD "umbrella" opens at an angle from ~100° to ~140° (Fig. 2b). The same hydrophobic residues (V139, I140, A142, V143, L145, V187, F193, and A195) are involved in the dimerization (Supplementary Fig. 9b).

The TM1-TM2 loop and the hinge region likely play an important role in the structural transition between the two forms, given that they define the relative arrangement of the helical bundles and BDs (Supplementary Movie 1). In the short form, the hinge region is inserted more deeply into the cavity formed at the top of the bundle (Fig. 2e). E115 in the hinge region is a key residue for structural transition. It forms a salt bridge with K50 (TM3) and a hydrogen bond with the main chain of A40 (TM1-2 loop) in the long form, whereas it forms a hydrogen bond with Y47 (TM3) in the short form. The hydrogen bond between the F112 carboxy group and the Y95 side chain is maintained in both forms. Notably, the hinge region and its key interaction residues Y47, K50, and Y95 are highly conserved in *Betacoronavirus*[21] (Supplementary Fig. 10). Moreover, a previous study on MHV M protein showed that the E121A/K/R mutations (corresponding to E115 in SARS-CoV-2 M protein) hinder virus and VLP formation, suggesting the importance of E115 in the hinge region for virus assembly[21]. In addition, lipid-like densities interacting with the hinge region, TM1, and TM1-2 loop were observed in the long form and resulted in the stabilization of this conformation (Supplementary Fig. 9c). These data indicate that the hinge region and related interactions are crucial for M protein function and that the structural transition between the two forms might be essential for virus assembly.

## M protein resembles ion channel ORF3a

The overall folding of the M protein was similar to the recently reported cryo-EM structure of SARS-CoV-2 ORF3a when searched using the Dali server[19]. The ORF3a of the *Betacoronavirus* subgenus *Sarbecovirus*, which includes SARS-CoV-2, has been reported as a non-specific cation-permeable channel[19,22,23]. Although the SARS-CoV-2 M protein and ORF3a share only 14.80% of their sequence identities (Supplementary Fig. 11), their structures are similar; the long form and short form structures of the M protein can be superposed on the ORF3a structure with root-mean-square deviation (RMSD) values of 3.7 and 4.4 Å, respectively (Fig. 3a). This structural similarity raises the possibility that the M protein also functions as viroporin.

To test this possibility, we examined the possible ion conduction pathways in the M protein. Both the long and short forms contain an upper cavity and a lower tunnel along the central dimerization interface, which are ~12-Å distances apart (Fig. 3b). The upper cavity, which is accessible from the intravirion side, is formed by the intravirion halves of TM1 and TM2, the TM1-2 loop, and the hinge region (Fig. 3b). Above the upper cavity the BD "umbrella" forms a vestibule to the cavity

(Fig. 3b). The entrances to the vestibule are formed by the edges of the BD (β2 and β5), the TM1-2 loop, and the hinge region (Fig. 3c). Multiple positive and negative residues, such as R42, R44, R131, E135, E137, and H155, are mapped to the entrance and the inside of the vestibule. The upper cavity of the short form is larger in volume than that of the long form (Fig. 3b, d). Y39 and W31 seal the bottom of the upper cavity in the long and short forms, respectively (Fig. 3b, d). The upper cavities are polar and partially embedded in the membrane, potentially incorporating water molecules and ions. The lower tunnel, connected to the extravironic space, is formed by only TM1 in the long form and by TM1 and TM2 in the short form (Fig. 3b). W31 and I24 seal the top of the lower tunnel in the long and short forms, respectively (Fig. 3d). Unlike the upper cavities, the lower tunnels are highly hydrophobic, with an average distance of ~6 Å between the residue pairs, making it unlikely that water molecules or ions could approach them.

Next, we conducted molecular dynamics (MD) simulations to investigate the possibility of ion conduction in M protein dimers. Consistent with the structural analysis, water molecules and ions could enter the upper cavities but not the lower tunnels in both forms of M protein (Fig. 3e and Supplementary Fig. 12). Therefore, ion conduction through the M protein was not observed during the simulations under any of the conditions examined. Thus, despite the structural similarities between the M protein and ORF3a, these data collectively suggest that it is unlikely that the M protein functions as an ion channel.

## M protein interacts with N protein and RNA through the basic residues

To gain mechanistic insight into M protein-mediated virus assembly, we studied the interactions among the M protein, the N protein, and RNA molecules. First, we utilized the liquid-liquid phase separation (LLPS) phenomenon of N protein[24]. N protein formed condensates to which M protein colocalized, indicating their interaction (Fig. 4a). Next, we examined the interaction of N and M proteins with poly(I:C) RNA. Both N and M proteins co-eluted with poly(I:C), suggesting that each protein directly binds to poly(I:C), although the poly(I:C) binding to the M protein was extremely weak (Supplementary Fig. 13). Furthermore, we examined the interaction between M and N proteins in the absence or presence of RNA by pull-down assays using FLAG-tagged M protein (Fig. 4b) and StrepII-tagged N protein (Fig. 4c). M protein was able to pull down N protein, and the interaction was enhanced in the presence of RNA (Fig. 4b), and vice versa (Fig. 4c).

Next, we investigated the region of the M protein responsible for the interaction with the N protein. We noticed that the intravirion surface of the M protein was highly basic (Fig. 4d). To determine whether the basic regions of the M protein are important for the interaction with the N protein, we constructed mutant M proteins (mutant #1–#7) (Fig. 4e). The basic regions could be spatially subdivided into four specific patches (Fig. 4d, e): mutant #1 contained residues of BD mapped to β6–β8; mutant #2 contained residues of BD mapped to the entrance of the upper cavity vestibule; mutant #3 contained residues in the juxtamembrane region spatially below mutants #1 and #2; and mutant #4 contained residues in the juxtamembrane region located in the TM1-2 loop near the entrance of the upper cavity vestibule. We then examined their binding ability to the N protein by immunoprecipitation assay (Fig. 4d, e). Two BD truncated forms (mutants #6 and #7) showed no binding to the N protein, demonstrating that BD is essential for recruiting N protein. No and weak binding to N protein were observed in mutant #2 and #4, respectively, whereas mutants #1 and #3 coprecipitated with the N protein. Thus, negatively charged residues mapped to the entrance of the upper cavity vestibule in the M protein are important for the recruitment of the N protein (Figs. 3c, 4d). The impaired binding of these mutants to the N protein might be caused by interference affecting direct interactions or inefficient RNA incorporation. Interestingly, the truncated form of the hinge region (mutant #5) still pulled

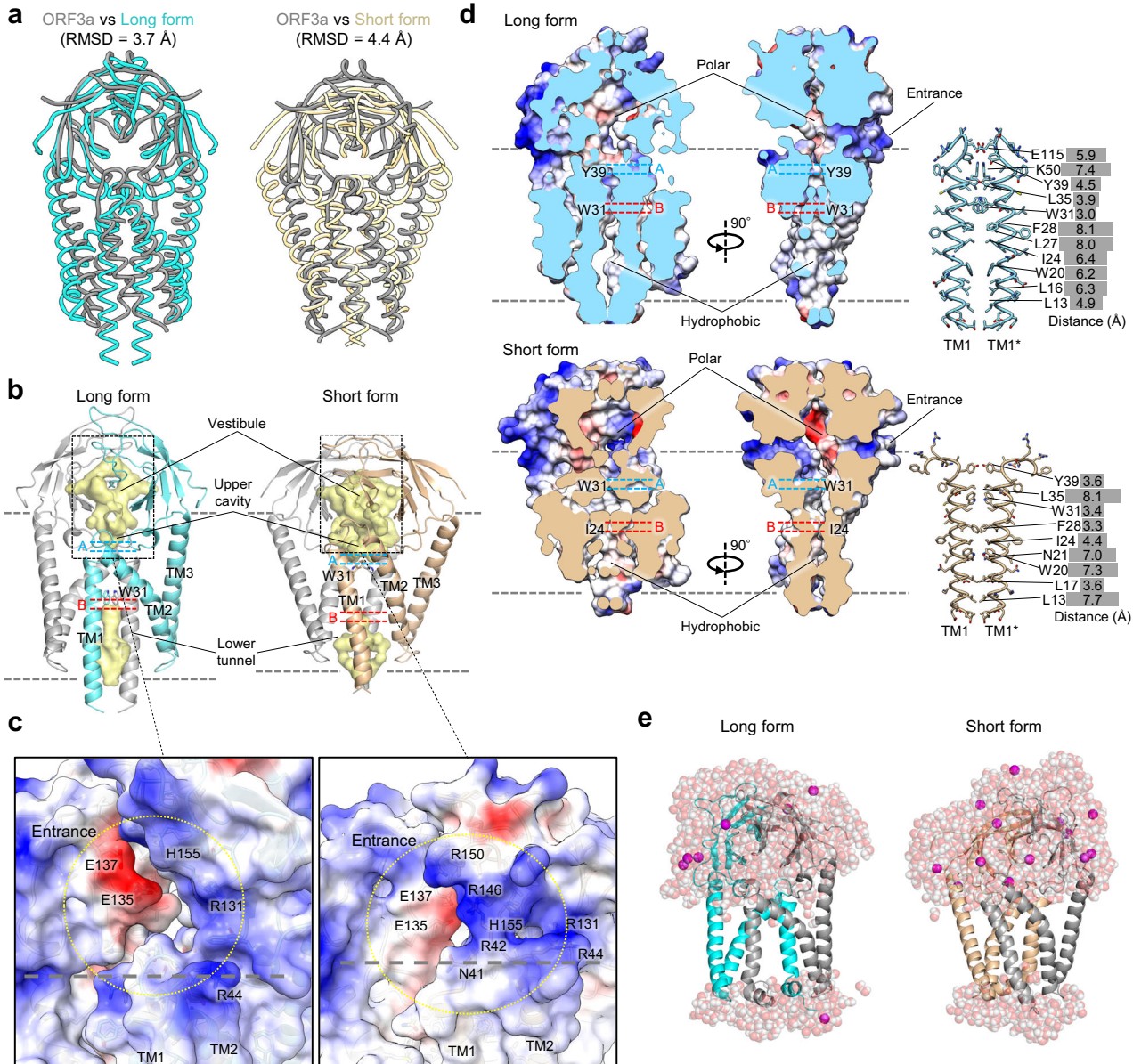

**Fig. 3 | The M protein resembles the ion channel ORF3a. a** Structural comparisons of the long form (cyan) and the short form (beige) of SARS-CoV-2 M protein dimer with SARS-CoV-2 ORF3a (gray) (PDB: 6XDC). **b** The inner surface of the M protein dimers (vestibule, upper cavity, and lower tunnel) calculated using Pymol are shown in yellow. The bottom of the upper cavities and the top of the lower tunnel are indicated as a, b, respectively. The W31 side chain is shown using a stick representation. **c** Close-up views of the entrances to the vestibules above the upper cavity. Half-transparent electrostatic surface potentials and side chains of charged residues are shown. The purple dashed lines indicate the approximate positions of the lipid bilayer. **d** Electrostatic surface potentials of the long and short forms and cross-sections along the C2 axis (left). Detailed views of the interface between TM1 and TM1* are shown, including the distances between each residue pair (right). **e** Snapshots of the MD simulation results of the long form (left) and the short form (right).

down the N protein, despite this region being extremely important for virus formation.

Taken together, we can conclude that the M protein itself weakly recruits the N protein and that the efficiency of this recruitment is synergistically enhanced by the co-existence of the N protein and RNA (Fig. 4f). Furthermore, the M protein mutagenesis assay demonstrated the critical role of the basic intravirion surfaces in the M protein for N protein recruitment.

## Discussion
Our work reveals the long-awaited structures of the SARS-CoV-2 M protein in two distinct conformations, providing opportunities to understand the mechanism of virus assembly. Previous low-resolution cryo-EM tomography studies have suggested the presence of two

conformations of MHV M protein structure inside the virion[20], which would correspond to the long and short forms of the M protein dimer identified in this study. We characterized the highly conserved hinge region, whose mutations and deletions inhibit virus formation[21], as a key motif that mediates conformational changes between the two forms (Fig. 2e). Since the deletion of the hinge region did not impair the interaction with the N protein (Fig. 4e), we conclude that the M–N interaction is not sufficient for virus assembly, and both forms of M protein are prerequisites in virus assembly. During the review process of this work, a preprint paper reported the structure of M protein dimer reconstituted in lipid nanodisc that corresponds to the short form in our study[25], which is highly complementary to our study and further strengthens our conclusion that both forms are important. Interestingly, we found lipid-like densities possibly stabilizing the long

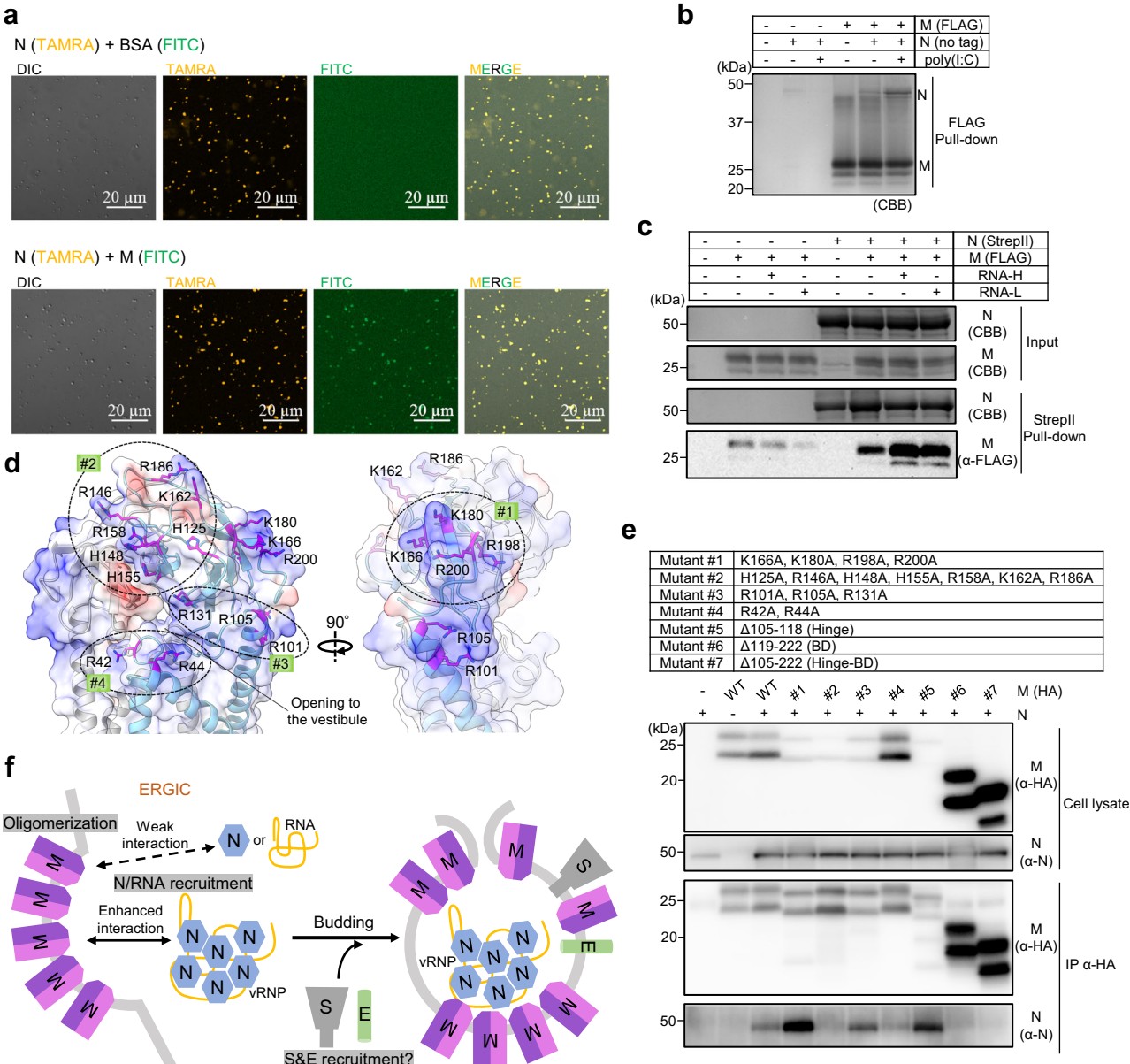

**Fig. 4 | M protein, N protein, and RNA interactions. a** Differential interference contrast (DIC) and fluorescence images of liquid-liquid phase separation (LLPS) of N protein (TAMRA-labeled) with or without M protein (FITC-labeled). The experiment was repeated three times with similar results. **b** FLAG tag pull-down assay using recombinant M protein (FLAG-tagged) and N protein (no tag) in the absence or presence of poly(I:C). The experiment was repeated twice with similar results. Source data are provided as a Source Data file. **c** StrepII tag pull-down assay using recombinant N protein (StrepII-tagged) and M protein (FLAG-tagged) in the absence or presence of RNAs of different sizes. RNA-H and RNA-L indicate yeast RNA with molecular weights >30 kDa and 3–30 kDa, respectively. The experiment was repeated twice with similar results. **d** Electrostatic surface potentials of the

intravirion side of the M protein dimer (long form). Positively charged residues to which the mutations were introduced in **e** are shown using stick representations and labeled. **e** Co-immunoprecipitation assay of wild-type (WT) or mutant M proteins with N protein in HEK293T cells. The experiment was repeated three times with similar results. Source data are provided as a Source Data file. **f** Model of M protein-triggered SARS-CoV-2 assembly M protein in the endoplasmic reticulum−Golgi intermediate compartment (ERGIC) forms dimers in two different conformations that assemble into higher-order oligomers to induce membrane curvature. M protein recruits N and genomic RNA in a cooperative manner. S and E proteins are also recruited to the budding site via an unknown mechanism. Source data are provided as a Source Data file.

form (Supplementary Fig. 9c). Considering that M protein-triggered coronavirus assembly takes place in the ERGIC-specific membrane environment and lipids are likely an important regulator for proper M protein conformation and function at this stage. Tomographic studies show that the long form is responsible for the rigidity and narrow curvature of the viral envelope and the recruitment of the S protein, while the short form provides more flexibility and lowers the spike density[20]. In this study, we observed tandemly arranged M protein oligomers that seemed to induce a slight curvature of the membrane (Fig. 1c). M protein oligomerization and the associated membrane

curvature can easily be propagated through the membrane, which might contribute to the formation of a round virion particle shape. Although a precise understanding of the mechanism by which the M protein induces membrane curvature and controls virus morphology requires future structural analysis of M protein oligomers, both forms of the M protein might be important for this process.

Five out of six monoclonal antibodies tested as fiducial markers in cryo-EM analyses were visualized in the 2D analyses, all of which showed binding to the intravirion side of the M protein (Supplementary Figs. 5–7). To our knowledge, no neutralizing antibodies have

been reported for SARS-CoV-2 targeting M protein. Because only a short stretch of the glycosylated N-terminal region is exposed on the extravirion side, it would be difficult to obtain a neutralizing antibody targeting M protein (Supplementary Fig. 10). The two Fab fragments (Fab-E and Fab-B) used in structure determination both recognized BD (Fig. 2a). The binding sites of both Fabs in the M protein span the dimerization interfaces of BD, which contain shared binding residues but show different structural features between the two forms of M protein dimer (Supplementary Fig. 14). Specifically, the binding sites for Fab-E contain residues mapped to the β2-β3 loop, β3–β4, β5, β6, and β7–β8, while the binding sites for Fab-B contain residues mapped to the hinge region, β1, β2, the β2–β3 loop, β6–β8, and the C-terminal loop (Supplementary Figs. 10, 14). Therefore, each Fab specifically recognizes and stabilizes each form of the M protein dimer. Antibodies recognizing the distinct conformational state of the M protein would be useful for studying M protein in terms of its structural dynamics. More importantly, the structures in which Fab-E or Fab-B specifically stabilize either form of M protein, raise the possibility of developing such therapeutic molecules to stabilize a certain conformation of M protein to block virus assembly. In this regard, the space between the transmembrane region and BD (i.e., the vestibule and upper cavity) identified in this study would offer an attractive target site (Fig. 3). Since M protein mutates rather slowly compared to S protein, therapeutic agents targeting M protein instead of S protein have the advantage of being less susceptible to mutations. For example, S proteins of SARS-CoV-2 (Uniport ID: P0DTC2) and Bat coronavirus RaTG13 (Uniport ID: A0A6B9WHD3) are slightly different in sequence (97.41% sequence identity), whereas M proteins of these two species are almost identical (99.55% sequence identity) (Supplementary Fig. 10). Similarly, only a minor mutation (I82T) in the M protein is known for lineage B.1.617.2 (Delta variant)[26]. Therefore, fixation of M protein conformation may be a promising approach to combat current and upcoming variants of SARS-CoV-2.

Ion channels commonly exist inside the virions of enveloped viruses; for example, the influenza virus M2 proton channel and the coronavirus E protein both have crucial functions in virus assembly and budding[27]. Although this study revealed structural similarities between the M protein and ORF3a (Fig. 3), it is unlikely that the M protein functions as an ion channel because its TM region is highly hydrophobic and has no apparent ion permeation pathway. However, we cannot exclude the possibility that the two known forms of the M protein represent the closed conformations and another unidentified form is responsible for ion conduction, or that M protein oligomers may function as ion channels. Further experimental research is needed to determine whether the M protein is an ion channel.

M protein in the ERGIC provides a scaffold for recruiting other viral structural proteins, such as N, E, and S proteins[10]. M and N are highly positively charged proteins. Our data demonstrate that the recruitment efficiency of N protein by M protein is synergistically enhanced by the presence of RNA, resulting in efficient packaging of viral RNA. Combining current knowledge on M proteins, we proposed a model for SARS-CoV-2 assembly (Fig. 4f). M protein dimers in the ERGIC might be in different conformations and oligomerize to induce membrane curvature. M protein also triggers the packaging of the genomic RNA into the virion by recruiting N protein and RNA in a cooperative manner, together with S and E proteins.

Our data pave the way for understanding M protein-triggered *Betacoronavirus* assembly mechanisms and provide a structural basis for developing virus assembly inhibitors.

## Methods
### Data reporting
No statistical methods were used to predetermine sample size. The experiments were not randomized, and the investigators were not blinded to allocation during experiments and outcome assessment.

### Preparation of recombinant SARS-CoV-2 membrane protein
The gene encoding the full-length SARS-CoV-2 membrane (M) protein (Uniprot: P0DTC5) codon-optimized for human cells was synthesized (Eurofins K.K., Japan). The DNA sequence (residues 1–222) with an N-terminal 8xHis-FLAG tag followed by a tobacco etch virus (TEV) protease recognition sequence (MHHHHHHHHDYKDDDDKENLYFQG) was cloned into pEZT-BM vector[28]. For expression, Expi293F cells (Thermo Fisher Scientific) cultured in Expi293 Expression Medium were transfected with the vector DNA/polyethylenimine (1 µg DNA per ml culture, w/w = 1:4) complex at a cell density of ~3.0 × 10$^6$ cells per ml and incubated at 37 °C under 8% CO$_2$ with agitation at 125 r.p.m. At 24-h post-transfection, the cell culture was supplemented with sodium butyrate at a final concentration of 5–10 mM and further incubated for 48 h at 30 °C. The cells were collected by centrifugation and were disrupted by sonication in a buffer containing 20 mM Tris-HCl, pH 7.5, 500 mM NaCl, and 5% (w/v) glycerol. After the removal of large insoluble debris by centrifugation at 2500×g for 10 min at 4 °C, the membrane fraction was collected by ultracentrifugation at 180,000×g for 1 h. The membrane fraction was resuspended into buffer A (10 mM Tris-HCl, pH 7.5, 250 mM NaCl, and 2.5% (w/v) glycerol) using a glass Dounce homogenizer and was solubilized by 1.0% LMNG (Anatrance) and 0.1% CHS (Anatrance) or 2% digitonin (Nacalai tesque) with gentle rotation for 1.5 h at 4 °C. The insoluble fraction was removed by centrifugation at 50,000×g for 10 min. The supernatant containing the solubilized protein was incubated with anti-DYKDDDDK tag antibody beads (Fujifilm) with gentle rotation for 1.5 h at 4 °C. The beads were washed with 20 column volumes of buffer A supplemented with 0.0025% LMNG and 0.00025% CHS for LMNG/CHS solubilized proteins or buffer A supplemented with 0.01% GDN (Anatrace) for digitonin solubilized proteins. The proteins were eluted by a buffer B (10 mM Tris-HCl, pH 7.5, 125 mM NaCl, 1.25% (w/v) glycerol, and 5 M LiCl) supplemented with 0.0025% LMNG, 0.00025% CHS for LMNG/CHS solubilized proteins or buffer B supplemented with 0.01% GDN for digitonin solubilized proteins. The elutes were concentrated using Amicon Ultra centrifugal filters (Merck, 100-kDa MW cut-off). LMNG/CHS solubilized proteins were incubated with TEV protease (home-made) for tag cleavage. Samples were further purified by gel filtration chromatography (Superose 6 Increase 10/300 GL, Cytiva) in 20 mM HEPES-NaOH, pH 7.6, 150 mM NaCl, 0.003% LMNG and 0.0003% CHS for LMNG/CHS solubilized proteins or 20 mM HEPES-NaOH, pH 7.6, 150 mM NaCl, and 0.01% GDN for digitonin solubilized proteins. The dimeric fractions or oligomeric fractions were concentrated to approximately 5–10 or 1.8 mg ml$^{-1}$, respectively, using Amicon Ultra centrifugal filters (Merck, 100-kDa MW cut-off). The purified proteins were flash-cooled in liquid nitrogen and stored at −70 °C until use.

### Mice and monoclonal antibody generation
All the animal experiments conformed to the guidelines of the Guide for the Care and Use of Laboratory Animals of Japan and were approved by the Kyoto University Animal Experimentation Committee. Mouse monoclonal antibodies against M protein were raised according to the previous method[29]. Briefly, a proteoliposome antigen was prepared by reconstituting purified M protein at high density into phospholipid vesicles consisting of a 10:1 mixture of egg phosphatidylcholine (Avanti Polar Lipids) and the adjuvant lipid A (Sigma) to facilitate immune response. BALB/c mice (female, 6 weeks of age, maintained at temperature and humidity ranges of 22 to 26 °C and 40 to 60% humidity, respectively, under a 12-h light, 12-h dark cycle.) were immunized with the proteoliposome antigen using three injections at 2-week intervals. Antibody-producing hybridoma cell lines were generated by a somatic fusion of B lymphocytes of the spleen with NS-1 myeloma cells (ATCC, cat# TIB-18) using a conventional fusion protocol. Hybridoma clones producing antibodies recognizing conformational epitopes in M protein were selected by an enzyme-linked immunosorbent assay on immobilized phospholipid vesicles

containing purified M protein (liposome ELISA), allowing positive selection of the antibodies that recognized the native conformation of M protein. Additional screening for reduced antibody binding to SDS-denatured M protein was used for negative selection against linear epitope-recognizing antibodies. Stable complex formation between M protein and each antibody clone was checked using fluorescence-detection size-exclusion chromatography. The sequences of the Fab from the antibodies were determined via standard 5′-RACE using total RNA isolated from hybridoma cells.

## Grid vitrification and cryo-EM data acquisition

For the LMNG/CHS solubilized M protein dimer without antibody, protein was diluted to 2.8 mg/mL, in a buffer containing 20 mM HEPES-NaOH, pH 7.7, 150 mM NaCl, 0.0025% LMNG, and 0.00025% CHS. For the GDN-solubilized M protein oligomer, purified protein solution at 1.8 mg ml$^{-1}$ was directly used for grid preparation. For the M/Fab-A (w/w = 1:1.9), M/Fab-B (w/w = 1:1.9), M/Fab-C (w/w = 1:1.1), M/Fab-D (w/w = 1:1.9), M/Fab-E (w/w = 1:1.9), or M/Fab-F (w/w = 1:1.2) complexes, LMNG/CHS solubilized M protein dimer was mixed with each Fab in a final protein concentration of 2.5, 1.5, 2.5, 1.5, 2.0, or 2.0 mg ml$^{-1}$, respectively, and incubated on ice for 30 min before grid preparation. A buffer containing 20 mM HEPES-NaOH, pH 7.6, 150 mM NaCl, 0.003% LMNG and 0.0003% CHS was used for sample dilution. The concentrations of the Fab stock solutions were in the range of 4–12 mg ml$^{-1}$.

For each sample, a 3-µl aliquot was applied onto a glow-discharged Quantifoil grid (R1.2/R1.3 300 mesh, copper), blotted for 4.5–5.5 s in 100% humidity at 4 °C, and plunged into liquid ethane using a VitrobotMkIV (Thermo Fisher Scientific). Cryo-EM micrographs were obtained by using a Titan Krios G3i microscope or a Titan Krios G4 microscope (Thermo Fisher Scientific) running at 300 kV and equipped with a Gatan Quantum-LS Energy Filter (GIF) and a Gatan K3 camera in the electron counting mode at the Cryo-EM facility in the University of Tokyo (Tokyo, Japan). Imaging was performed at a nominal magnification of 105,000×, corresponding to a calibrated pixel size of 0.83 Å/pixel. For large dataset collections, each movie was recorded for 3.0 s and subdivided into 60 frames with an accumulated exposure of 57–58 e$^-$/Å$^2$ at the specimen (Supplementary Figs. 2, 3) or 5.0 seconds and subdivided into 64 frames with an accumulated exposure of 61–62 e$^-$/Å$^2$ at the specimen in the correlative double sampling (CDS) mode (Supplementary Figs. 4, 5). Movies were acquired by the image shift method using the SerialEM software[30], with defocus ranges of −1.2 to −2.2 µm or −0.8 to −1.6 µm (CDS mode). For small to medium dataset collections (Supplementary Fig. 7), each movie was recorded for 2.5 s and subdivided into 50 frames with an accumulated exposure of about 50 e$^-$/Å$^2$ at the specimen. Movies were acquired by fast acquisition mode using the EPU software (Thermo Fisher Scientific) with a defocus range of −1.0 to −2.2 µm.

## Cryo-EM data processing and model building

Image processing was conducted using RELION-3.1[31,32] and/or cryoSPARC v3.0.0-3.2.0[33]. The dataset of LMNG/CHS solubilized M protein dimer without antibody was processed using cryoSPARC (Supplementary Fig. 2). 6,055 raw movie stacks were motion-corrected using the patch motion correction, and the CTF parameters were determined using the patch CTF estimation. 5918 micrographs were selected based on their CTF resolution and a total of 9,464,727 particles were picked using the template picker. After multi rounds of 2D classification, ab initio reconstruction and heterogeneous refinement, a package containing 348,142 particles were obtained, which yielded a 6.2-Å resolution 3D reconstruction using the non-uniform (NU) refinement[34]. The dataset of GDN-solubilized M protein oligomer was processed using cryoSPARC (Supplementary Fig. 3). About 3195 raw movie stacks were processed similarly. 1,267,328 particles were picked and cleaned by multi rounds of 2D classification. Subsequent ab initio

reconstruction, heterogeneous refinement, and NU refinement generated only very low-resolution 3D reconstruction.

The dataset of the M/Fab-E complex was processed using cryoSPARC (Supplementary Fig. 4). About 5562 raw movie stacks collected in CDS mode were processed similarly in cryoSPARC[33]. A subset containing 696 micrographs were used to generate 2D templates and 3D volume references. A total of 3,992,290 particles were initially picked using the template picker. After removal of 2D classes with no Fab binding, two rounds of heterogeneous refinement yielded a package of 263,166 particles, which yielded a 2.7-Å resolution 3D reconstruction using the NU refinement. The 3D map was sharpened by applying a B-factor of −50.0 Å$^2$. The local resolution map was produced with the local resolution estimation program in cryoSPARC[33].

For the M/Fab-B complex (Supplementary Fig. 5), firstly, a test dataset containing about 600 micrographs collected using EPU software was used to generate 2D templates for particle picking and initial 3D reference map for 3D classification. Mainly, 3539 raw movie stacks collected in CDS mode were motion-corrected using the RELION implemented version of MotionCor2[35]. The CTF parameters were determined using the CTFFIND4 program[36]. Particle picking, multi rounds of 3D classification, 3D auto-refine, Ctf refinement and bayesian polishing in RELION[32] finally yielded 22,011 particles, which yielded a 2.8-Å resolution 3D reconstruction using NU refinement in cryoSPARC. The 3D map was sharpened by applying a B-factor of −40.0 Å$^2$. The local resolution map was produced with the local resolution estimation program in cryoSPARC[33].

The other datasets of the M/Fab-A (66 movie stacks), M/Fab-C (59 movie stacks), M/Fab-D (551 movie stacks), and M/Fab-F (102 movie stacks) complexes were processed using cryoSPARC[33] and 2D classification analysis was conducted (Supplementary Fig. 7).

For model building of the M/Fab-E complex, M protein was manually built using COOT[37]. The Fab-E molecules were built by using the crystal structure of anti-Ghrelin receptor antibody (PDB: 6KS2)[38] as an initial model, which was fitted into the 3D map using Chimera[39] and then manually refined using COOT[37]. For model building of the M/Fab-B complex, the BD and the three-helix bundle of the M protein (long form) from the M/Fab-E complex were separately fitted into the 3D map using Chimera and manually refined in COOT[37]. Fab-B molecules were built similarly to Fab-E. Real-space refinement in Phenix program[40] was performed after manual refinement. The cryo-EM maps have been deposited in the Electron Microscopy Data Bank. The atomic coordinates of the M/Fab-E and M/Fab-B complex structures have been deposited in the Protein Data Bank. Statistics for data collection and structural refinement are summarized in Supplementary Table 1. Structure representations were generated using Pymol[41], Chimera[39], or ChimeraX[42].

## Preparation of recombinant SARS-CoV-2 nucleocapsid protein

The cDNA encoding SARS-CoV-2 nucleocapsid (N) protein (Cat# VG40588-NH) was purchased from Sino Biological. The amino acid sequence is identical to the NCBI reference sequence YP_009724397.2, except for a point mutation: G335A. The DNA sequences (residues 2-419, G335A) with either an N-terminal 6xHis-FLAG tag followed by an HRV-3C protease recognition sequence (MSYYHHHHHHDYKDDDDK-LEVLFQGPEF) or an N-terminal 6xHis-StrepII tag followed by an HRV-3C protease recognition sequence (MSYYHHHHHHWSHPQFEK-LEVLFQGPEF) were cloned into pFastBac Dual vector (Life Technologies). Recombinant baculovirus were generated using ExpiSf9 cells (Thermo Fisher Scientific). For expression, ExpiSf9 cells were infected with 5% (v/v) passage-3 baculovirus. After incubation at 27 °C for 60 h, cells were collected, resuspended, and lysed by sonication in a buffer containing 25 mM Tris-HCl, pH 8.0, 500 mM NaCl, 5% (w/v) glycerol, 20 mM imidazole pH 8.0, and 1% protease inhibitor cocktail (Nacalai tesque). Cleared lysate by centrifugation was supplemented with 0.02% Triton X-100 and 100 mM NaCl. The proteins were then purified

by Ni-NTA resin (Fujifilm). For FLAG-tagged N protein, tags were cleaved by HRV-3C protease (home-made). The proteins were further purified by Superdex 200 gel filtration column (GE Healthcare). The proteins were concentrated to about 10–20 mg ml$^{-1}$ in a buffer containing 20 mM HEPES-NaOH, pH 7.5, and 500 mM NaCl using Amicon Ultra centrifugal filters (Merck, 50-kDa MW cut-off). The purified proteins were flash-cooled in liquid nitrogen and stored at −70 °C until use.

## LLPS assay and microscope

For fluorescent dye labeling of N protein, 300 μg purified N protein (tag cleaved) was mixed with 0.2 mM 5-carboxytetramethylrhodamine (TAMRA) *N*-succinimidyl ester (TCI, Cat# T2808) in a final volume of 300 μL and incubated on ice for 1 h. After incubation, Tris-HCl, pH 8.0 was added to a final concentration of 50 mM to quench the reaction. Excess dye was removed by passing through a HiTrap Desalting column (Cytiva) equilibrated with a buffer containing 10 mM Tris-HCl, pH 8.0, and 500 mM NaCl. For fluorescent dye labeling of M protein, 100 μg LMNG/CHS solubilized M protein dimer was mixed with 0.1 mM Fluorescein (FITC)−5-maleimide (TCI, Cat# F0810) in a final volume of 100 μL and incubated on ice for 30 min. Then the buffer was exchanged to a buffer containing 10 mM HEPES-NaOH, pH 7.7, 150 mM NaCl, 0.003% LMNG, and 0.0003% CHS using Amicon Ultra centrifugal filters (Merck, 100-kDa MW cut-off). FITC-labeled bovine serum albumin (BSA, Nacalai tesque) was prepared similarly to FITC-labeled M protein. Labeled proteins were flash-cooled in liquid nitrogen and stored at −70 °C until use.

The LLPS of N protein and M protein was observed using a ZEISS LSM 800 confocal microscope with a 40× water immersion objective in differential interference contrast (DIC) and fluorescent imaging modes. The samples contained 40 μM TAMRA-labeled N protein (10% labeled), 18 μM FITC-labeled M protein (or 16 μM FITC-labeled BSA), 10 mM HEPES-NaOH, pH 7.7, 65 mM NaCl, 0.003% LMNG, and 0.0003% CHS. A 10-μL aliquot was placed on a sliding glass (Matsunami Glass) and covered with a round cover glass (Matsunami Glass). Images were taken from freshly prepared (within 10 min) samples.

## Size-exclusion chromatography analysis

For the binding assay of M protein and Fab (Supplementary Fig. 1c), LMNG/CHS solubilized M protein dimer (10 μg) and/or Fab (20 μg) were diluted to a volume of 60 μL and subjected to size-exclusion chromatography (SEC) using Superdex 200 Increase 5/150 GL column (GE Healthcare) in a buffer containing 10 mM Tris-HCl, pH 7.5, 150 mM NaCl, 0.003%LMNG, and 0.0003% CHS at a flow rate of 0.5 mL/min. For the binding assay of N protein and poly(I:C) (Supplementary Fig. 13a), N protein (tag cleaved, 150 μg) and/or poly(I:C) (Invivogen, Cat# tlrl-picw) (90 μg) were diluted to a volume of 150 μL and subjected to SEC using a repacked Superdex 200 Increase 10/300 GL column (GE Healthcare) in a buffer containing 10 mM HEPES-NaOH, pH 8.0, 100 mM NaCl, and 0.02% *N*-Octyl-β-D-glucopyranoside at a flow rate of 1.0 mL/min. Eluted fractions were analyzed by SDS-polyacrylamide gel electrophoresis (SDS-PAGE) stained with Coomassie Brilliant Blue (CBB). For the binding assay of M protein and poly(I:C) (Supplementary Fig. 13b), GDN-solubilized M protein dimer (10 μg) with or without poly(I:C) (40 μg) were diluted to a volume of 50 μL and subjected to SEC using a repacked Superdex 200 Increase 10/300 GL column (GE Healthcare) in a buffer containing 10 mM HEPES-NaOH, pH 8.0, 100 mM NaCl, and 0.01% GDN at a flow rate of 1.0 mL/min. Eluted fractions were analyzed by dot blot detected by primary antibody: Anti-DDDDK-tag mAb (MBL, Cat# M185-3L, Lot# 015, dilution 1:2,000) and secondary antibody: Rabbit Anti-Mouse IgG H&L (HRP) (Abcam, Cat# ab6728, Lot# GR3383345-1, dilution 1:2,000); chemiluminescence reagent: Chemi-Lumi One, Nacalai tesque).

All samples were monitored for absorbance at 280 and 260 nM. SEC data were visualized using Matplotlib.

## Pull-down assay

For FLAG tag pull-down (Fig. 4b), N protein (tag cleaved, 150 μg) was mixed with or without GDN-solubilized M protein dimer (14 μg) in the presence or absence of poly(I:C) (90 μg) in buffer C (10 mM HEPES-NaOH, pH 8.0, 100 mM NaCl, and 0.02% GDN) (total volume of 130 μL). Five microliters of the solution was analyzed by SDS-PAGE as input. Then, 45 μL (net 15 μL) anti-DYKDDDDK tag antibody beads (Fujifilm) equilibrized in buffer C was added to each sample and incubated with gentle mixing for 1 h at 4 °C. The resin was washed three times with 1 mL buffer C. The proteins were eluted in 60 μL of a buffer containing 100 mM Glycine-HCl, pH 3.0, 100 mM NaCl, and 0.02% GDN. An 18-μL aliquot of each eluate was analyzed by SDS-PAGE stained with CBB.

Yeast RNA (Sigma, Cat# R6625-25G) was dissolved in a buffer containing 10 mM Tris-HCl, pH 8.0, and 0.5 mM EDTA-K to a concentration of about 10 mg/mL and was fractionated according to the size of RNA by using Amicon Ultra centrifugal filters (Merck, 30-kDa MW cut-off and 3-kDa MW cut-off), which gave two fractions named as RNA-H (>30 kDa) and RNA-L (3–30 kDa). For StrepII tag pull-down, GDN-solubilized M protein dimer (12 μg) was mixed with or without StrepII-tagged N protein (50 μg) in the presence or absence of RNA-H or RNA-L in buffer D (10 mM HEPES-NaOH, pH 7.5, 50 mM NaCl, and 0.01% GDN) (total volume of 200 μL). Ten microliters of the solution was analyzed by SDS-PAGE as input. Then, 30 μL (net 12 μL) Strep-Tactin®XT 4Flow® resin (IBA) equilibrized in buffer D was added to each sample and incubated with gentle mixing for 2 h at 4 °C. The resin was washed three times with 1 mL buffer D. The proteins were eluted in 60 μL of a buffer containing 10 mM HEPES-NaOH, pH 7.5, 500 mM NaCl, 100 mM biotin, and 0.01% GDN. For detection of N protein, an 18-μL aliquot of each eluate was analyzed by SDS-PAGE stained with CBB. For detection of M protein, a 10-μL aliquot of each eluate was analyzed by western blotting (antibody: Anti-DDDDK-tag mAb-HRP-DirecT, MBL, Cat# M185-7, Lot# 009, dilution 1:2,000; chemiluminescence reagent: Chemi-Lumi One, Nacalai tesque).

## Mutagenesis assay

To generate M mutants #1 (K166A, K180A, R198A, R200A), #2 (H125A, R146A, H148A, H155A, R158A, K162A, R186A), and #3 (R101A, R105A, R131A), DNA fragments of the M gene containing respective mutations were synthesized (Eurofins K.K., Japan), and the DNA fragments were amplified by PCR with appropriate primers. Using In-Fusion HD Cloning Kit (Takara bio), the PCR products were ligated with inverse PCR products amplified from the plasmid expressing N-terminally HA-tagged M protein (pCAGGS/HA-M). The M mutant #4 (R42A, R44A) and deletion mutants #5 (Δ105-118), #6 (Δ119-222), and #7 (Δ105-222) were generated by PCR-based site-directed mutagenesis and inverse PCR with specific primers using pCAGGS/HA-M as a template, respectively. All constructs were sequenced to ensure that unwanted mutations were not present.

For immunoprecipitation assay, 293 T cells, which were maintained in Dulbecco's modified Eagles medium containing 10% fetal bovine serum, were transfected with plasmids expressing HA-M, HA-M mutants, and/or N. At 22 h after transfection, the cells were lysed in lysis buffer (1%Triton X-100, 0.5% sodium deoxycholate, and 0.1% SDS in PBS) and centrifuged at 15,000×*g* for 10 min. The supernatants were immunoprecipitated by using mouse anti-HA-tag antibody (HA-probe (F-7); sc-7392, Santa Cruz Biotechnology) and Protein G coupled with magnetic beads (Dynabeads protein G for Immunoprecipitation, Thermo Fisher Scientific). Immunoprecipitants were eluted by boiling with 2X SDS sample buffer and analyzed by western blotting. Rabbit polyclonal antibody for SARS-CoV-2 nucleocapsid protein (GTX135357, GeneTex, dilution 1:10,000) and secondary antibodies, anti-rabbit IgG (HRP) (NA934, GE Healthcare, dilution 1:10,000), were used for detecting N protein. Also, mouse monoclonal antibody against HA-tag [HA-probe (F-7); sc-7392, Santa Cruz Biotechnology, dilution 1:10,000] and anti-mouse Ig (HRP) (Mouse TrueBlot Ultra; 18-8817-30, Rockland

Immunochemicals, dilution 1:10,000) were used for detecting M protein.

## Molecular dynamics simulation

Initial models of the short and long forms were prepared for MD simulations from the M/Fab complex structures solved by cryo-EM. Two Fab fragments were deleted. To align the length of the M protein in the short and long forms, the 205 and 206 residues at the C-terminal of the long form were deleted; the 9–204 residues of the M protein were used. The orientation of the M protein relative to the lipid bilayer was set to that of the ORF3a protein deposited in the Orientations of Proteins in Membranes (OPM) database (ID: 7KJR)[43]. The cryo-EM structure of the M protein was aligned to the structure of the ORF3a protein in the OPM database, and the membrane position was in good agreement with that observed in the cryo-EM data. The short and long forms of the M protein were embedded in 1-palmitoyl-2-oleoyl-sn-glycero-3-phosphocholine (POPC) membranes and water molecules using the Membrane Builder implemented in CHRARMM-GUI[44–49]. The missing hydrogen atoms of the M protein were added, and the protonation states were assigned using PROPKA[50,51] implemented in PDB2PQR[52] under pH 7 conditions. E167 was set as a protonated glutamic acid. The N- and C-termini were set as $NH_3^+$ and $COO^-$, respectively. A rectangular unit cell was prepared. In the center of the unit cell, the M protein was embedded in an ~70 × 70 Å$^2$ POPC bilayer along the x–y plane, and the numbers of POPC molecules in the upper and lower leaflets were ~60 and ~55, respectively. Along the z-axis of the unit cell, the cell was filled with water molecules (TIP3 water model[53]). Finally, 150 mM potassium (K$^+$) and chloride (Cl$^-$) ions, including their counterions, were added: The numbers of K$^+$ and Cl$^-$ ions were 29 and 51 for the short form, and 34 and 56 for the long form. The developed systems are shown in Supplementary Fig. 12a, b.

All-atom MD simulations were conducted using the MD program package GROMACS ver. 2019.6 under periodic boundary conditions[54–56]. The CHARMM36m force field[57–61] with the WYF parameter[62] for cation-pi interactions was used. According to the default setup in the Membrane Builder, energy minimization and six equilibration runs were performed before the production run, and their calculation conditions were the same as those of our previous simulations[63]. After the equilibration runs, a 2-μs production run was performed with the NPT ensemble in each short or long form system. The temperature and pressure were set at 300 K and 1 atm. The thermostat and barostat followed the Nosé–Hoover scheme[64,65] and the semi-isotropic Parrinello–Rahman approach[66,67], respectively. The electrostatic interaction was handled by the smooth particle mesh Ewald method[68]. The van der Waals interaction was smoothly truncated using the switching function within a range of 10–12 Å. Bond lengths involving hydrogen atoms were constrained by the P-LINKS algorithm[69], and the time step was set to 2 fs.

To mimic previous electrophysiological experiments on the ORF3a protein[19], all-atom MD simulations with an external electric field were conducted. Using the membrane–water M protein system described above, the electric field was applied along the z-axis, and 1-μs production runs were performed for the short and long forms separately. The strength of the electric field was set to 0.01 V/nm, corresponding to the ~0.1 V which was applied in the ORF3a protein experiments. The length of the unit cell along the z-axis was ~12 nm for the short form and ~13 nm for the long form and the electric field corresponding to 100 mV was estimated to be ~0.0082 V/nm and ~0.0075 V/nm for short and long forms, respectively. The 0.01 V/nm electric field used in the MD simulations was a slightly stronger value than that used in the experiments. After the equilibration runs described above, a 1-μs production run was performed with the electric field. During the 1-μs run, snapshots were extracted every 1 ns (1000 snapshots), and the z-axis positions of ions were analyzed.

## Reporting summary

Further information on research design is available in the Nature Research Reporting Summary linked to this article.

## Data availability

All data needed to evaluate the conclusions in the paper are present in the paper and Supplementary Information. Data and resources supporting this paper may be requested from the authors. The cryo-EM density maps have been deposited in the Electron Microscopy Data Bank (EMDB) under accession code EMD-31977 (M/Fab-E), EMD-31978 (M/Fab-B), and EMD-31976 (low-resolution M dimer). The coordinates have been in the RCSB Protein Data Bank (PDB) under accession code 7VGR (M/Fab-E) and 7VGS (M/Fab-B). For referenced structures, the cryo-EM structure of SARS-CoV-2 ORF3a and structure of anti-Ghrelin receptor antibody under accession codes PDB: 6XDC and PDB: 6KS2, respectively. Source data are provided with this paper.

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

## Acknowledgements

We thank Masahide Kikkawa, Haruaki Yanagisawa, Akihisa Tsutsumi, Yoichi Sakamaki, and Yoshiaki Kise for the management and support of the Graduate School of Medicine cryo-EM facility at the University of Tokyo, and Chiho Onishi for the support of plasmid construction. This work was supported by a Grant-in-Aid from the Japanese Ministry of Education, Culture, Sports, Science, and Technology Grant Nos. 20K15730 and 22K15046 (Z.Z.), 19H03164 and 22H02556 (U.O.), 19H00976 and 21K19328 (T.S.), the "Program for Promoting Researches on the Supercomputer Fugaku" (MD-driven Precision Medicine) (project ID: hp200129 and hp210172 to M.I.), and by a Grant-in-Aid for Scientific Research on Innovative Areas "Molecular Engine" (grant no: 18H05426 to M.I.); CREST, JST (JPMJCR21E4) (T.S.); Sumitomo Dainippon Pharma Co., Ltd. (T.S.); the Takeda Science Foundation (U.O. and T.S.); the Mochida Memorial Foundation for Medical and Pharmaceutical Research (U.O.); the Daiichi Sankyo Foundation of Life Science (U.O.); and the Naito Foundation (U.O.); the Research Foundation for Pharmaceutical Sciences (Z.Z.); the Basis for Supporting Innovative Drug Discovery and Life Science Research (BINDS) from the Japan Agency of Medical Research and Development (AMED) (Grand No. JP21am0101115; support No. 1570, 1846, 1848; Grant No. 21am0101079; N.N. and S.I.; Grant No. JP22ama121023 to M.I); and the Research on Development of New Drugs from the AMED (N.N. and S.I.); the JST CREST (JPMJCR20HA) (T.N.); the JSPS Core-to-Core Program A; the Advanced Research Networks (T.N.); the Joint Usage/Research Center program of Institute for Frontier Life and Medical Sciences Kyoto University (N.N. and T.N). The grant for 2021-2023 Strategic Research Promotion (No. SK202202 to M.I.) of Yokohama City University.

## Author contributions

Conceptualization: Z.Z.; Investigation (recombinant protein generation): Z.Z., and M.Y.; Investigation (cryo-EM analysis and image processing): Z.Z. and U.O.; Investigation (antibody generation): N.N., T.U., K.L., and S.I.; Investigation (mutagenesis assay): Y.M. and T.N.; Investigation (pull-down assay): Z.Z.; Investigation (LLPS assay and microscope): Z.Z., N.K., and J.A.; Investigation (MD simulation): T.E. and M.I.; Visualization: Z.Z.; Validation, data curation and project administration: Z.Z. and U.O.; Writing (original draft): Z.Z. and U.O.; Writing (review & editing): Z.Z., U.O., and T.S.; Supervision: U.O. and T.S.; Funding acquisition: Z.Z., U.O., and T.S.

## Competing interests

The authors declare no competing interests.
