## [Peer Review File · Nature Communications]

Structure of SARS-CoV-2 membrane protein essential for virus assemblyReviewers' Comments:

Reviewer #1:

Remarks to the Author:

SARS-CoV-2 mainly consists of four structural proteins: spike (S) protein, nucleocapsid (N) protein, envelope (E) protein, and membrane (M) protein. There are already a lot of studies focus on the spike protein, the authors choose a relatively unpopular target, M protein. They tried different Fabs and solved two high-resolution structures of M-protein in different conformation called long-form and short-form. With validation from the low-resolution oligomerized M protein structure without Fab and biochemical studies, the authors illustrated that the role of M protein for virus assembly and other possible functions of M protein such as working as ion channel. However, the evidence for making hypothesis and conclusion needs further verified.

Specific concerns:

1. It can be hardly draw conclusion from the 2D image (Fig.1) that membrane curved caused by oligomerization. At least from my eyes, I could not observe obvious bending of the micelle. The low-resolution density including micelles couldn't show the curved shape neither. The conclusion should be made with more caution.
2. Fig 4d. is difficult to read. The presentation of the mutant residues' region is crowded and needs to be highlighted.
3. The density presentation of ED Fig.6 (Hinge, BD, Fab) is overlapped and too crowded, and it is difficult validate the quality of the map.
4. The interaction details such as the related residues and the binding distance could be highlighted compared with those residues not involved as important as the key residues. (eg. The ED Fig.9 hydrophobic residues discussed in main text)
5. In Fig. 3d, some labeling is not clear. (Polar)
6. The conformational difference between long-form and short-form M protein is one of evidence proved by the previous studies and fit with non-Fab bound dimer structure. However, the discussion about their biological function on the virion and how Fab B or Fab E could structurally stabilized M protein maintain in specific conformation is not enough.
7. The M protein is under the same condition before binding with different Fabs, M protein should be equilibrium between long-form and short-form then stabilized to one form by Fab. But the low resolution non-Fab bound dimer structure only matches one of them. Is there any 3D classification of non-Fab data could show short-form matched density?
8. In Fig.3 and discussion, author discussed the possibility of M protein function as ion channel. However, as mentioned in discussion, there was no further experimental evidence but only sequence analysis and structural comparison. The hypothesis and analysis based on the cavity volume and channel distance may not be convincing. If the biochemical assay proving its ion channel function were difficult, the computational simulation could provide some information to indicate the ion pathway.
9. Furthermore, since the structures of ORF3a and M protein show high similarity in SARS-CoV-2, the essentiality of M protein working as ion channel might need to be discuss. The relationship between virus assembly by sequence alignment and the ion channel from structural similarity need further explanation.
10. "No and weak binding to N protein were observed in mutant #2 and #4, respectively, whereas

mutants #1 and #3 coprecipitated with the N protein”

The description of the mutants #1- #4 should in more detail in main text. Such as describe the reason of residues selection in these mutants.

11. The description of buffer should be unified. The buffer A and B in pulldown assay and purification are different.

12. “Multiple positive and negative residues, such as R42, R44, R131, E135, E137, and H155, are mapped to the entrance to and inside the vestibule.” The “to” after entrance should be deleted.

Reviewer #2:

Remarks to the Author:

This is a significant study that has resolved the structure of the SARS-CoV-2 M protein by cryo electron microscopy. The results are very impactful as they inform of two different M conformations, short and long, that have been previously proposed (although not resolved) for SARS-CoV-1 years ago. The structure of M helps shed light on the mechanism by which SARS-CoV-2 may assemble and bud from the host ERGIC compartment. The authors do a great job interpreting the structure and discussing it in the context of not only SARS-CoV-2 literature but that of previous coronavirus work. I find the paper to be well positioned in the field, objective and not over interpreted.

Methods are all sound and appropriate replicates are included. Careful details are included for other scientists to attempt and repeat the work.

I only have one strong recommendation for writing revision. The authors do a good job of M-N Co-IP experiments of M and N interactions for WT and mutants of M. However, I'd temper the interpretation of direct M residue electrostatics in binding N. Indeed, it could be the case but without more experiments mutation of cationic residues in M could also influence other M residues that are needed for direct N interactions. This is a minor point but an important one for the final paper and interpretation.

Response to reviewers

The comments of all reviewers were very useful in helping us to improve the quality of our manuscript. Our responses to the comments of each reviewer can be found below.

The Figures and Table have been reorganized as follows:

Original	Revised
Fig. 3e, f	Removed Fig. 3e newly created
Fig. 4d	Modified
Ext. Data Fig. 6	Modified
Ext. Data Fig. 9	Modified Ext. Data Fig. 12 newly created
Ext. Data Fig. 12	Ext. Data Fig. 13
Ext. Data Fig. 13	Moved to Ext. Data Fig. 14 and panel c added

Reviewer #1 (Remarks to the Author):

SARS-CoV-2 mainly consists of four structural proteins: spike (S) protein, nucleocapsid (N) protein, envelope (E) protein, and membrane (M) protein. There are already a lot of studies focus on the spike protein, the authors choose a relatively unpopular target, M protein. They tried different Fabs and solved two high-resolution structures of M-protein in different conformation called long-form and short-form. With validation from the low-resolution oligomerized M protein structure without Fab and biochemical studies, the authors illustrated that the role of M protein for virus assembly and other possible functions of M protein such as working as ion channel. However, the evidence for making hypothesis and conclusion needs further verified.

Response: We appreciate the careful consideration of our manuscript by reviewer #1. Please find our point-by-point responses below. We believe that all the comments and suggestions helped to improve the manuscript.

Specific concerns:

1. It can be hardly draw conclusion from the 2D image (Fig.1) that membrane curved caused by oligomerization. At least from my eyes, I could not observe obvious bending of the micelle. The low-resolution density including micelles couldn't show the curved shape neither. The

conclusion should be made with more caution.

Response: As the reviewer pointed out, we agree that it is difficult to conclude the membrane curvature only from the 2D images and that careful conclusion should be made. Therefore, this subjective conclusion about membrane curvature has been removed from the result part (page 6, line 12 in the original manuscript) and moved to the discussion part with a weaker argument.

(page 15, line 6)

“The tomography studies show that the long form is responsible for the rigidity and narrow curvature of viral envelope and recruitment of S protein, while the short form brings more flexibility and low spike density (ref#20). In this study, we observed tandemly arranged M protein oligomers were likely to induce a slight curvature of the membrane (Fig. 1c). M protein oligomerization and associated membrane curvature can be easily propagated through the membrane, which might contribute to the formation of a round virion particle shape. Although precise understanding the mechanism by which M protein induces membrane curvature and controls virus morphology requires future structural analysis of M protein oligomers, both of the two forms of M protein might be important for this process.”

2. Fig 4d. is difficult to read. The presentation of the mutant residues' region is crowded and needs to be highlighted.

Response: We improved Fig 4d by highlighting the relevant residues in magenta to make it easier to read.

3. The density presentation of ED Fig.6 (Hinge, BD, Fab) is overlapped and too crowded, and it is difficult to validate the quality of the map.

Response: We improved extended data Fig.6 by displaying the subdivisions of the structures. We think the current version is easier to validate the map quality.

4. The interaction details such as the related residues and the binding distance could be highlighted compared with those residues not involved as important as the key residues. (eg. The ED Fig.9 hydrophobic residues discussed in main text)

Response: According to the reviewer's suggestion, we listed the key residues with contact distances less than 4.5 Å and generated the interaction maps (dimerization interface within

a single bundle of the TM region and between the BD-BD interface) in extended data Fig.9.

5. In Fig. 3d, some labeling is not clear. (Polar)

Response: Thank the reviewer for pointing it out. The problem might be due to some bugs during pdf conversion. We updated these blurred labels.

6. The conformational difference between long-form and short-form M protein is one of evidence proved by the previous studies and fit with non-Fab bound dimer structure. However, the discussion about their biological function on the virion and how Fab B or Fab E could structurally stabilized M protein maintain in specific conformation is not enough.

Response: Related to the comment#1, we added more discussion about the biological function of the two forms in the discussion part with reference to the previous tomography study.

(page 15, line 6)

“The tomography studies show that the long form is responsible for the rigidity and narrow curvature of viral envelope and recruitment of S protein, while the short form brings more flexibility and low spike density (ref#20). In this study, we observed tandemly arranged M protein oligomers were likely to induce a slight curvature of the membrane (Fig. 1c). M protein oligomerization and associated membrane curvature can be easily propagated through the membrane, which might contribute to the formation of a round virion particle shape. Although precise understanding the mechanism by which M protein induces membrane curvature and controls virus morphology requires future structural analysis of M protein oligomers, both of the two forms of M protein might be important for this process.”

Besides, in the result part, we add more detailed recognition mechanisms of M protein by FabE and FabB as follows.

(page 16, line 3)

“The two Fab fragments (Fab-E and Fab-B) used in structure determination both recognized BD (Fig. 2a). The binding sites in M protein for both Fabs span the dimerization interfaces of BD, which contain shared binding residues but show different structural features between the two forms of M protein dimer (Extended Data Fig. 14). Specifically, binding sites for Fab-E contains residues mapped to β 2- β 3 loop, β 3~ β 4, β 5, β 6, β 7~ β 8, while binding sites for Fab-B contains residues mapped to the hinge region, β 1, β 2, β 2- β 3 loop, β 6~ β 8 and the C-

terminal loop (Extended Data Fig. 10 and 14). Therefore, each Fab specifically recognizes and stabilizes each form of the M protein dimer. “

7. The M protein is under the same condition before binding with different Fabs, M protein should be equilibrium between long-form and short-form then stabilized to one form by Fab. But the low resolution non-Fab bound dimer structure only matches one of them. Is there any 3D classification of non-Fab data could show short-form matched density?

Response: We did not observe the 3D class that matched the short-form in the cryo-EM dataset of the non-Fab bound M protein dimer (Extended Data Fig. 8). In this revision, we re-analyzed the cryo-EM dataset with more carefulness, but again, we could not identify any short forms, only long form.

Since we describe “M protein is in a conformational equilibrium between the two states, and that each state was captured by its conformation specific antibody (page 7, line 11)”, it is natural for the reviewer to be concerned that the short form was not observed. We are, of course, aware of this point as well.

It should be noted that not all structures in the sample can be visualized by cryo-EM analysis. Especially in the case of small molecular size proteins that are difficult to analyze with cryo-EM, as in this case, it is not surprising that even if there is a short form, it may not be resolved depending on the number of particles and their flexibility. In fact, the previous tomography study (ref#20) has shown a high percentage of the long form in virus, therefore it is possible that the percentage of the long form is also high in M protein samples solubilized with detergent.

Although the presence of the short form could not be confirmed in the cryo-EM analysis of the detergent-solubilized Fab unbound M-protein sample (Extended Data Fig. 8), the fact that Fab-B, which specifically recognizes the short form, was obtained using M-protein reconstituted in liposomes as antigen, is a strong evidence that the short form is present in the lipid environment and is one of the physiologically relevant forms of M protein.

8. In Fig.3 and discussion, author discussed the possibility of M protein function as ion channel. However, as mentioned in discussion, there was no further experimental evidence but only sequence analysis and structural comparison. The hypothesis and analysis based on the cavity volume and channel distance may not be convincing. If the biochemical assay proving its ion channel function were difficult, the computational simulation could provide some information to indicate the ion pathway.

Response: As the reviewer mentioned, experimental or computational evidence is vital for investigating whether M protein functions as an ion channel. Unfortunately, we could not conduct the experimental assay such as patch clamp due to equipment limitations. Instead, according to the reviewer's suggestion, we performed MD simulations to investigate the possibility of ion conduction of M protein.

The simulations were performed with several conditions (with or without electric field) using the long and short forms of M proteins as starting model. Unfortunately, no ion conduction (potassium or chloride ion) was observed under all conditions tested using either the long-form or the short-form structure of M protein dimer (please see figures below). Consistent with the structural analysis, ions and water molecules can enter the upper cavity, However, the hydrophobic property of the lower tunnel precludes the ion conduction.

Long form

Short form

Revised Fig. 3e

Snapshots of the MD simulation results of the long form (left) and the short form (right).

Revised Extended Data Fig. 12

Initial models of M protein for MD simulations and trajectories of ions

a, b, Initial membrane-water system of M protein for short form (a) and long form (b). M protein dimer is shown as cartoon representations, POPC membrane is represented by line

representations, and water molecules, potassium, and chloride ions are represented by sphere representations for cyan, yellow, and magenta color, respectively. The z-axis is defined so that the axis is perpendicular to the membrane surface. c-f, Trajectories of ions along z-axis during MD simulations with an external electric field. The averaged positions of phosphorus atoms in membrane molecules of the upper and lower leaflets are cyan plots. Other plots are the positions of potassium and chloride ions during the MD simulation of short form (c, d) and long form (e, f).

Based on the results of the MD simulation, we now conclude that the M protein dimer is less likely to function as an ion channel. Therefore, the claim “M protein is a putative ion channel” in the original manuscript should be weakened in the revised manuscript. Accordingly, in the revised manuscript, we have removed the descriptions that assume ion conduction through M protein. Moreover, we described the results of the MD simulation, and concluded that M protein is unlikely to function as an ion channel in the present structures.

(page 12, line 1)

Next, we conducted the MD simulations to investigate the possibility of ion conduction in M protein dimers. Consistent with the structural analysis, water molecules and ions could enter the upper cavities. However, they could not pass (penetrate?) the lower tunnels in both forms of M protein (Fig. 3e). Therefore, ion conduction through M protein was not observed during the simulations under all conditions examined. Collectively, despite structural similarity of M protein to ORF3a, M protein unlikely functions as an ion channel.

However, we could not exclude the possibility that an unidentified dimer form other than the two forms observed in this study or oligomer form of M protein function as ion channel. Therefore, we mention this possibility in the discussion section as follows,

(page 17, line 10)

Although this study revealed the structural similarity between M protein and ORF3a (Fig. 3), it is unlikely that M protein functions as an ion channel because its TM region is highly hydrophobic and has no apparent ion permeation pathway. However, we can not exclude the possibility that both of the two forms of M protein represent the closed conformations and another unidentified form is responsible for ion conduction or that M protein oligomers function as ion channel. Further experimental research is needed to determine whether M protein is an ion channel.

Although the description concerning the putative ion channel function has now been substantially changed, we believe that the main conclusions and purpose of the manuscript are consistent.

9. Furthermore, since the structures of ORF3a and M protein show high similarity in SARS-CoV-2, the essentiality of M protein working as ion channel might need to be discussed. The relationship between virus assembly by sequence alignment and the ion channel from structural similarity need further explanation.

Response: Since the original claim that “M protein is a putative ion channel” has now been changed in the revised version in response to the previous comment, we could not respond to this reviewer’s comment. As stated above, the hydrophobicity of the TM region of M protein may account for the difference in ion channel function between the M protein and ORF3a.

10. “No and weak binding to N protein were observed in mutant #2 and #4, respectively, whereas mutants #1 and #3 coprecipitated with the N protein”

The description of the mutants #1- #4 should in more detail in main text. Such as describe the reason of residues selection in these mutants.

Response: We added more descriptions about these mutants in the manuscript, including the reason for selecting these mutants.

(page 13, line 6)

“The basic regions could be spatially subdivided into four specific patches (Fig. 4d, e): mutant #1 contains residues of BD mapped to the $\beta 6$ - $\beta 8$; mutant #2 contains residues of BD mapped to the entrance of the upper cavity vestibule; mutant #3 contains residues in the juxtamembrane region spatially below mutant #1 and #2; mutant #4 contains residues in the juxtamembrane region located in the TM1-2 loop near the entrance of the upper cavity vestibule.”

11. The description of buffer should be unified. The buffer A and B in pulldown assay and purification are different.

Response: We renamed the buffers for more clarity. Specifically, buffer A and B in the pull-down section were renamed buffer C and D, respectively.

12. “Multiple positive and negative residues, such as R42, R44, R131, E135, E137, and H155,

are mapped to the entrance to and inside the vestibule.” The “to” after entrance should be deleted.

Response: This “to” is now deleted.

Reviewer #2 (Remarks to the Author):

This is a significant study that has resolved the structure of the SARS-CoV-2 M protein by cryo electron microscopy. The results are very impactful as they inform of two different M conformations, short and long, that have been previously proposed (although not resolved) for SARS-CoV-1 years ago. The structure of M helps shed light on the mechanism by which SARS-CoV-2 may assemble and bud from the host ERGIC compartment. The authors do a great job interpreting the structure and discussing it in the context of not only SARS-CoV-2 literature but that of previous coronavirus work. I find the paper to be well positioned in the field, objective and not over interpreted.

Methods are all sound and appropriate replicates are included. Careful details are included for other scientists to attempt and repeat the work.

Response: We thank this reviewer for the high recognition of our manuscript.

I only have one strong recommendation for writing revision. The authors do a good job of M-N Co-IP experiments of M and N interactions for WT and mutants of M. However, I'd temper the interpretation of direct M residue electrostatics in binding N. Indeed, it could be the case but without more experiments mutation of cationic residues in M could also influence other M residues that are needed for direct N interactions. This is a minor point but an important one for the final paper and interpretation.

Response: We appreciate the thoughtful suggestion by the reviewer. The direct interaction between M protein and N protein was demonstrated by the LLPS and pull-down assays, and the interaction was enhanced in the presence of RNA in the pull-down assay. Therefore, as pointed out by the reviewer, the interpretation of the loss of binding to N protein observed in mutants #2/#4 could be other than just a effect on direct binding.

Following the reviewer's suggestion, we updated the conclusions as follows:

(page 13, line 16)

“Thus, negatively charged residues mapped to the entrance of the upper cavity vestibule in the M protein are important for the recruitment of the N protein (Fig. 3c, 4d). The impaired binding of these mutants to N protein might be due to interfering with direct interaction or inefficient incorporation of RNA.”

(page 14, line 4)

“Taken together, we can conclude that the M protein itself weakly recruits N protein and that this efficiency is synergistically enhanced by the co-existence of N protein and RNA (Fig. 4f). Furthermore, the mutagenesis assay of M protein indicated the critical role of the intravirion basic surfaces in M protein for N protein recruitment.”

Reviewers' Comments:

Reviewer #1:

Remarks to the Author:

In the revised version of manuscript and response, the authors improved their manuscripts and answered most of concerns. The displaying and the logic of story are clear enough. With the additional MD simulation results, even the results indicating opposite conclusion to authors previous hypothesis of ion-channel like function, the question of M protein function could be open for more possibility based on the further exploration. I have no other concerns, just a minor suggestion: at figure legend of Fig3 in page 20 line 12 and the title in page 10 line 9, the ORF of ORF3a should be capitalized.

I think this work reached the sufficient level of publication.

Reviewer #2:

Remarks to the Author:

I find the revised work to be improved and a strong addition to the literature. The structural information, interpretation and data analysis is robust to come to confident conclusions on how M may function.

Reviewer #3:

Remarks to the Author:

Understanding SARS-CoV-2 M protein structure and structure-function relations are highly desirable. The authors present a very detailed structural characterization and analysis of this protein and add support to several possible functions. The structures reported (short and long forms) will help understand the viral assembly process. The structure of SARS-CoV-2 M protein by cryo-electron microscopy is an important addition to the tools available to address this pandemic, properly emphasized by the analysis presented. The presentation is well written in its revised form, and the authors address the concerns of the previous reviewers in great detail, which covers many of my concerns as well, including the excessive emphasis on the channel hypothesis in the original version, which I believe is now corrected. A paper was recently released in biorxiv reporting a similar structure [1]. The authors may want to acknowledge this publication with the proper clarification that it was published during the review process. The authors should also consider mentioning Ouzounis analysis [2], given the importance of the ORF3a/M structural similarity.

[1] "Structure of SARS-CoV-2 M protein in lipid", Dolan, K.A. et al.,
doi: <https://doi.org/10.1101/2022.06.12.495841>; June 13, 2022

[2] "A recent origin of Orf3a from M protein across the coronavirus lineage arising by sharp divergence", Christos A. Ouzounis, Computational and Structural Biotechnology Journal 18 (2020) 4093–4102

Response to reviewers

Reviewer #1 (Remarks to the Author):

In the revised version of manuscript and response, the authors improved their manuscripts and answered most of concerns. The displaying and the logic of story are clear enough. With the additional MD simulation results, even the results indicating opposite conclusion to authors previous hypothesis of ion-channel like function, the question of M protein function could be open for more possibility based on the further exploration. I have no other concerns, just a minor suggestion: at figure legend of Fig3 in page 20 line 12 and the title in page 10 line 9, the ORF of ORF3a should be capitalized.

Thank you for your supportive comments. According to the suggestion, we capitalized all ORF3a now.

I think this work reached the sufficient level of publication.

Reviewer #2 (Remarks to the Author):

I find the revised work to be improved and a strong addition to the literature. The structural information, interpretation and data analysis is robust to come to confident conclusions on how M may function.

We appreciate the reviewer's positive opinions.

Reviewer #3 (Remarks to the Author):

Understanding SARS-CoV-2 M protein structure and structure-function relations are highly desirable. The authors present a very detailed structural characterization and analysis of this protein and add support to several possible functions. The structures reported (short and long forms) will help understand the viral assembly process. The structure of SARS-CoV-2 M protein by cryo-electron microscopy is an important addition to the tools available to address this pandemic, properly emphasized by the analysis presented. The presentation is well written in its revised form, and the authors address the concerns of the previous reviewers in great detail, which covers many of my concerns as well, including the excessive emphasis on the channel hypothesis in the original version, which I believe is now corrected. A paper was recently released in biorxiv reporting a similar structure [1]. The authors may

want to acknowledge this publication with the proper clarification that it was published during the review process. The authors should also consider mentioning Ouzounis analysis [2], given the importance of the ORF3a/M structural similarity.

[1] "Structure of SARS-CoV-2 M protein in lipid", Dolan, K.A. et al., doi: <https://doi.org/10.1101/2022.06.12.495841>; June 13, 2022

[2] "A recent origin of Orf3a from M protein across the coronavirus lineage arising by sharp divergence", Christos A. Ouzounis, Computational and Structural Biotechnology Journal 18 (2020) 4093–4102

We thank the reviewer for the careful consideration and the proper recognition of our manuscripts. According to the suggestion, the biorxiv paper (ref#26) and the work of Ouzounis et al. (ref#18) are now acknowledged in the manuscript. Especially for the biorxiv paper, we added some discussion as follows:

(Page15, Line3-6)

During the review process of this work, a preprint paper reported the structure of M protein dimer reconstituted in lipid nanodisc that corresponds to the short form in our study (ref#26), which is highly complementary to our study and further strengthens our conclusion that both forms are important.